

# Detection of railway catenary insulator defects based on improved YOLOv5s

Jing Tang[1,2], Minghui Yu[2] and Minghu Wu[1,2]

[1] Hubei Key Laboratory for High-efficiency Utilization of Solar Energy and Operation Control of Energy Storage System, Hubei University of Technology, Wuhan, China
[2] School of Electrical and Electronic Engineering, Hubei University of Technology, Wuhan, China

## ABSTRACT

In this article, a method of railway catenary insulator defects detection is proposed, named RCID-YOLOv5s. In order to improve the network's ability to detect defects in railway catenary insulators, a small object detection layer is introduced into the network model. Moreover, the Triplet Attention (TA) module is introduced into the network model, which pays more attention to the information on the defective parts of the railway catenary insulator. Furthermore, the pruning operations are performed on the network model to reduce the computational complexity. Finally, by comparing with the original YOLOv5s model, experiment results show that the average precision (AP) of the proposed RCID-YOLOv5s is highest at 98.0%, which can be used to detect defects in railway catenary insulators accurately.

## INTRODUCTION

A catenary is essential for electric railways to obtain electric energy, and it is crucial to the normal and stable operation of electric railways. The catenary is a special form of transmission line installed on the railway line to supply power to electric locomotives. It is usually exposed to the air and vulnerable to the high-speed impact of electric locomotive pantographs, which implies that the catenary has become the weak link in the power supply system of the electrified railways. As part of the catenary, insulators are not only exposed to the atmospheric environment but also subjected to a strong electric field and strong mechanical tension. However, faults, such as broken and lost insulator pieces, are prone to occur, resulting in reduced insulation strength and thus threatening the operation of the electrified railway. More and more electrified railway accidents are caused by insulator failure, which implies that it is important to detect the defects of insulators. The existing detection methods are mainly based on manual inspection, supplemented by traditional image processing methods. However, manual inspection is time-consuming and cannot guarantee the accuracy. Thus, it is necessary to adopt an efficient and accurate detection method to detect the insulators of the railway catenary.

Recently, the combination of machine learning based classifiers and feature extractors has been frequently used for object detection in many fields including railroad systems. In 2001, Viola and Jones proposed an object detection method using the AdaBoost cascade

Corresponding author
Minghu Wu, 2217852364@qq.com

framework to classify each Haar-like feature (*Viola & Jones, 2001*). This method mainly applies classifiers trained on different types of features to calculate the position of the object and classify it in a sliding window. Many feature extractors have been widely used, including SIFT (*Lowe, 2004*), SURF (*Bay et al., 2008*), Haar-like, HOG (*Dalal & Triggs, 2005*), LBP (*Ojala, Pietikainen & Maenpaa, 2002*), and so on. *Wei et al. (2019)* utilized Sense-SIFT and BOVM to extract local features, while spatial features were extracted using spatial pyramid decomposition. Then, the extracted features were used to train the SVM classifier to finally achieve defect detection of railway fasteners. *Zhao & Liu (2014)* employed SURF to extract features of the insulators from images and used the intuitionistic fuzzy set (IFS) to divide the extracted features into k classes. Then, all the connected regions for each class were localized by rectangular boxes to accurately locate the insulators in the image. *Zhuang et al. (2018)* used extended Haar-like features to construct the features of rail cracks, then used a cascade classifier based on the LogitBoost algorithm to detect rail cracks. *Fan, Hou & Li (2018)* proposed a line LBP encoding method to detect the failed railway track fastener. However, the object detection methods based on a combination of classifiers and feature extractors are complex and time-consuming. However, object detection methods based on deep learning have attracted much attention due to their excellent generalization and high detection efficiency.

Most deep learning based object detection models are based on CNN (convolutional neural networks), including Fast R-CNN (region-based convolutional neural network) (*Girshick, 2015*), Mask R-CNN (*He et al., 2017*), YOLO (You Only Look Once) (*Redmon et al., 2016*), SSD (Single Shot MultiBox Detector) (*Liu et al., 2016*), and so on. *Liu et al. (2020b)* proposed an improved Faster R-CNN method to detect the brace sleeve screws, which used discrimination maps with high resolution and proposal maps with low resolution to identify and locate the brace sleeve screws. *Wu et al. (2021)* divided the detection of the catenary components into two parts. In the first part, a cascaded YOLO was used to locate the components, which integrate deep and shallow information. Then, in the second part, a rotation RetinaNet was used to detect the located components. Moreover, the defect detection of railway catenary components is generally divided into two parts, first locating the object, then detecting defects in the located object. For instance, *Zhao et al. (2021)* combined Faster R-CNN with FPN to locate the insulators from images, then detected the defects in the railway catenary insulators. *Kang et al. (2019)* used Faster R-CNN to localize the insulators and then proposed a deep multitask neural network to identify and classify the defects. *Sadykova et al. (2020)* employed YOLOv2 network model to detect and classify insulators for satisfying the real-time demand for insulator detection. *Liu et al. (2021)* improved the detection accuracy of insulator defects by adding an SPP module and a multi-scale feature fusion structure to YOLOv3-tiny. However, the method of detecting insulator defects in two parts is time-consuming. Due to the relatively large size of insulators in the components of the catenary, the object detection methods can be considered to detect the catenary insulator defects directly to reduce the time consumption.

In this article, the object detection method is used to directly detect the insulator defects in the images collected by the inspection vehicle. This operation eliminates the step of locating the insulator and greatly reduces the time consumption. The insulator defects

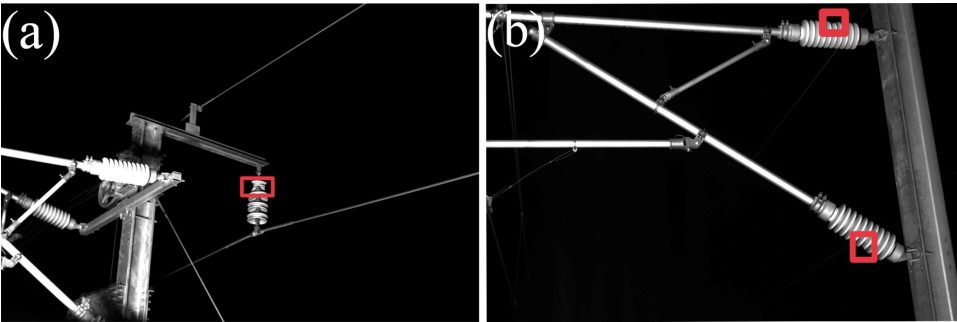

**Figure 1** Defects of the railway catenary insulator.

detected in this research refer to the breakage of insulators. Defects of the railway catenary insulator are shown in Fig. 1. In MS-COCO (*Lin et al., 2014*), the small object is defined as less than 32 × 32 pixels, this definition is affected by the size of the original image. In addition, there is a common definition that the object in an image is less than 1% of the whole image (*Liu et al., 2020a*). Moreover, according to the second definition (*Liu et al., 2020a*), the defects in railway catenary insulators can be treated as small objects, and because the inspection vehicle usually works at night, the overall background of the photographed images is relatively simple. However, the images collected at night are presented in black and white, which leads to some interference in the detection of insulator defects by other nearby railway catenary components. Hence, the proposed method of Railway Catenary Insulator Defects-YOLOv5s (RCID-YOLOv5s) in this article can enhance the ability to classify and locate railway catenary insulator defects on the premise of ensuring the detection speed.

The main contributions of this article are summarized as follows. Firstly, the defects of railway catenary insulators are detected and located directly, eliminating the step of separating the insulators from the image, thus substantially improving the detection speed of the network model. Secondly, since the defects of railway catenary insulators are not easily detected in the captured images, a small object detection layer is added to the network model to enhance the network's ability to detect small objects. Furthermore, the pruning operation is carried out on the network model to reduce the computational complexity and improve the detection speed.

The remainder of this article is organized as follows. 'Materials and Methods' describes the RCID-YOLOv5s method. The dataset and experimental results are described in 'Results'. Finally, the 'Conclusion' section presents the conclusion of the full text.

## MATERIALS AND METHODS

In this section, the detailed descriptions of the RCID-YOLOv5s method are outlined. Firstly, the overall structure of RCID-YOLOv5s is given as follows.

### Overall structure of RCID-YOLOv5s

The structure of the proposed RCID-YOLOv5s is shown in Fig. 2, which is based on the original YOLOv5s (*Glenn, 2020*) architecture. There are three parts of RCID-YOLOv5s:

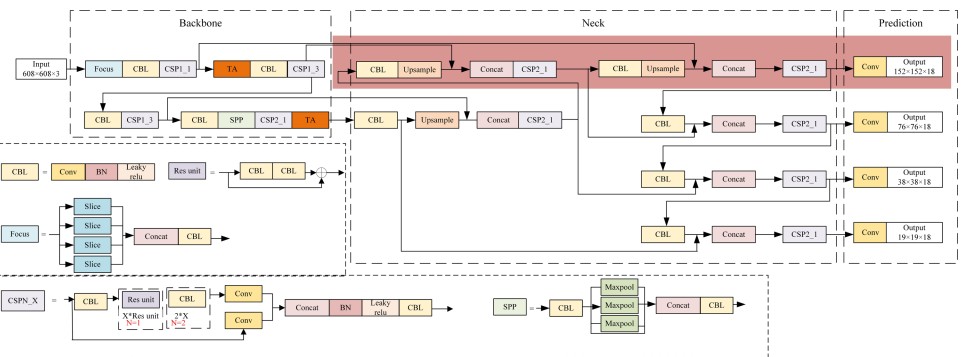

**Figure 2  The structure of RCID-YOLOv5s.**

backbone, neck, and prediction, like YOLOv5s. The backbone is used to extract features from the image. The neck is used to fuse the initial output features from the backbone and adapt the size of feature maps to improve the overall performance of the network model. The prediction of the network is used to receive the four outputs from the neck, and finally output the position of the prediction box and the confidence and category of the object. In the backbone, a focus module slices the input image to increase the speed and reduce the floating point operations of the network. The CBL module is a convolutional module for feature extraction. Two structures of CSP (*Wang et al., 2020*) modules are applied to the backbone and neck, respectively, to improve the learning ability of the network. The SPP (*He et al., 2015*) module is used to achieve the fusion of local features and global features to improve the representation ability of feature maps. The triplet attention (TA) (*Misra, Nalamada & Hou, 2021*) module is added to the backbone to assign different weights to different features, in which effective features are assigned more weights to reduce the impact of invalid features on the network detection results. The structure of triplet attention is shown in Fig. 3. The neck combines the FPN (feature pyramid network) (*Lin et al., 2017*) structure and the PAN (path aggregation network) (*Liu et al., 2018*) structure to fuse the initial output features from the backbone and pass them to prediction. The structure of FPN+PAN is shown in Fig. 4. The prediction receives the feature maps generated by neck and uses them to detect objects of different sizes, and finally outputs the position, confidence, and category of the objects. Meanwhile, the network model uses the CIoU function to calculate the regression loss and the binary cross-entropy function to calculate the confidence loss and classification loss.

In the following subsection, the demonstration of triplet attention is given.

## Triplet attention

The images used in this article to train the network model are collected by the inspection vehicle. Moreover, there is a large amount of feature information in these images that are not effective in improving the accuracy of the insulator defect detection model. Thus, it is necessary to assign different weights to the different feature information when training the network model. However, the original network model is trained by assigning the same

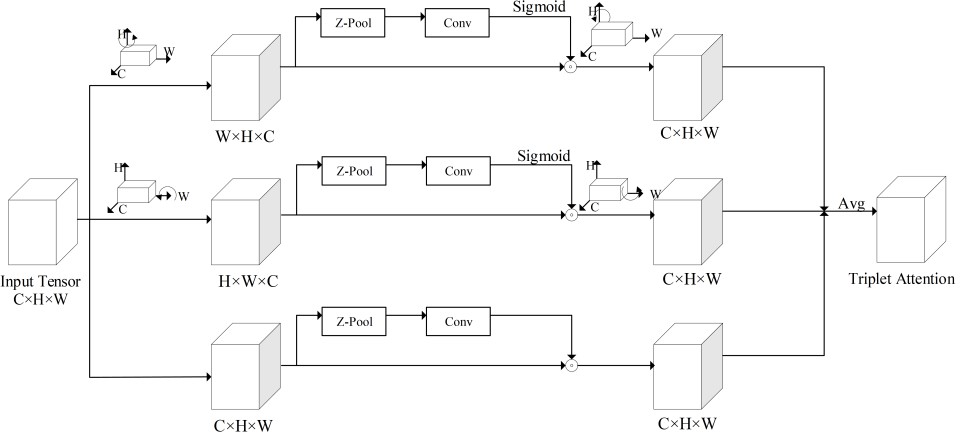

**Figure 3** The structure of triplet attention.

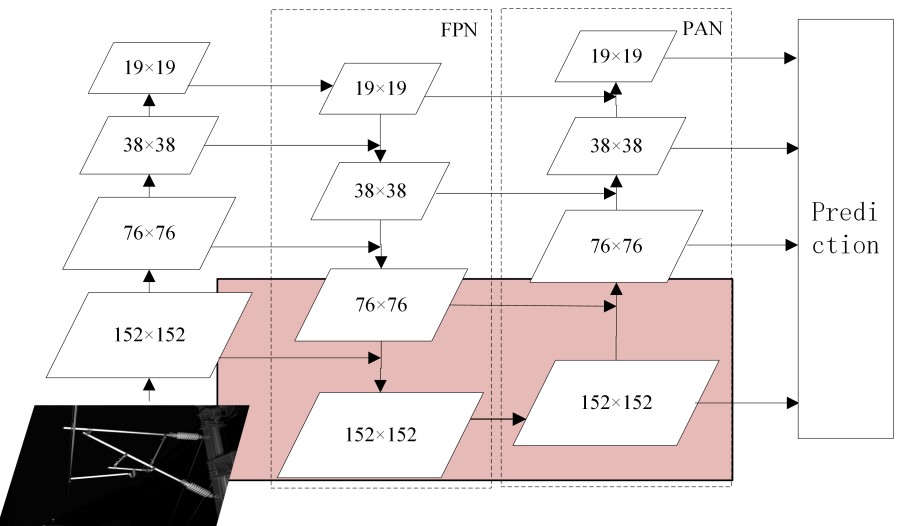

**Figure 4** The structure of FPN+PAN.

weights to all feature information. It means that there is no difference in weight assignment between valid and invalid feature information.

In this article, a TA module is introduced in the backbone of the network model, which can make the network model pay more attention to the valid feature information. The structure of triplet attention is shown in Fig. 3.

As shown in Fig. 3, the TA module is structured with three parallel branches, the correlations between two spatial dimensions and channel dimension are extracted by the first and second branches, and the spatial feature interdependencies are extracted by the third branch. In the first and second branches, a 90° anti-clockwise rotation is respectively performed in the input tensor, along the H and W axes. In the third branch, the shape of the input tensor is kept constant. Firstly, the tensor is reduced to two dimensions by

connecting the maximum pooling feature and the average pooling feature of the tensor through a Z-pool layer, which allows the layer to reduce the depth of the actual tensor while preserving its rich representation and without significantly increasing the computational complexity. Secondly, the reduced tensor is passed into a convolutional layer with a K × K convolutional kernel. Finally, the corresponding attention weights are generated by the Sigmoid activation function and added to the rotation tensor. In the final output, the output tensor is first rotated to the same shape as the input tensor in each of the first two branches, then summed equally with the output tensor of the third branch as the output of the TA model.

TA model can model both spatial attention and channel attention to achieve cross-dimensional information interaction, giving more weight to effective spatial features and channel features. The network model with TA module added can pay more attention to important feature information, which can reduce the influence of useless feature information on the network model. Moreover, it enhances the feature extraction ability of the network model at a small memory cost. Thus, by using the TA module, the ability of the network model to extract features can be improved and as well as the accuracy of detection.

Next, we introduce the small object detection layer (SODL).

## Small object detection layer (SODL)

The neck part of RCID-YOLOv5s combines FPN with PAN to fuse feature information from different detection layers in a bidirectional loop. The semantic information from the top layer is transferred to the bottom layer by the FPN structure, while the location information from the bottom layer is transferred to the top layer through the PAN structure. Moreover, the feature map for detection containing rich semantic and location information is obtained. This structure enables the network model to detect and localize objects more accurately.

The structure of FPN+PAN is shown in Fig. 4. For the detection image with an input size of 608 × 608, the original network model detects the three feature maps with sizes of 19 × 19, 38 × 38, and 76 × 76 output by the Neck part respectively, and outputs the location, category, and confidence of the object.

A feature map of size 76 × 76 cannot accurately locate the small object detection in this article. Thus, a detection layer for small objects is added to the original network model. A larger feature map of size 152 × 152 is generated in the neck part to enable the network model to better locate the small object. As shown in the pink part of Fig. 4, the feature map is further downsampled and upsampled based on the original network model and connected with the feature map of size 152 × 152 in backbone, after which a feature map of size 152 × 152 is generated for detecting small objects. The smaller the size of the feature map, the larger the area corresponding to each grid cell in the input image, as well as the richer semantic information of the object contained in the feature map. Therefore, the detection of a feature map of size 152 × 152 can locate smaller objects more accurately.

## Model pruning

To conclude, this subsection gives the model pruning of our article.

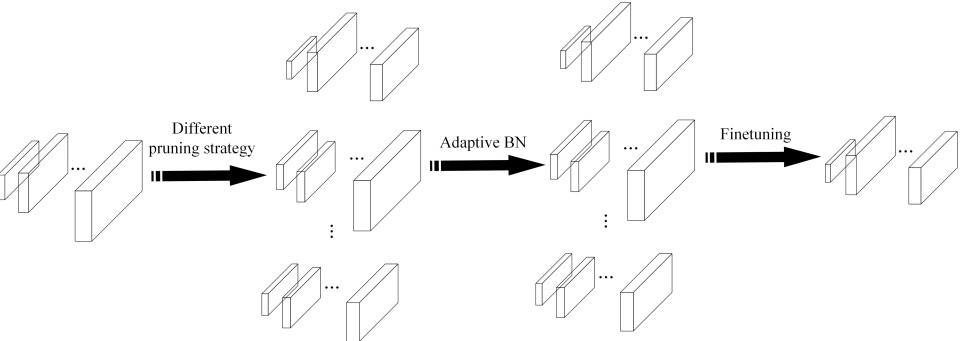

**Figure 5** **The EagleEye pruning process.**

The railway system is working all day and a failure could affect the normal operation of many trains and threaten the personal safety of passengers and staff. Therefore, the defect detection of railway catenary insulators is required to be completed in the shortest possible time. However, the increase in the number of network model layers could influence the detection speed. In this case, a method of model pruning is used to ensure the detection speed of the network model while improving the detection accuracy of the network model.

By evaluating the importance of some parameters or channels in the network model and removing some unimportant parameters or channels with minimal impact, the model pruning can reduce the size of the network model and speed up the detection. The general procedure of model pruning is to pre-train the model and prune the trained model. The pruning is not a one-time success but keeps pruning different layers and parameters of the original model, and a fine-tuning of all the pruned models is required. The purpose of fine-tuning is to minimize the loss of accuracy and select a pruned model that meets the desired object as the final network model.

To reduce the time consumption of model pruning operation, the EagleEye (*Li et al., 2020*) pruning method is used to prune the network model, which can determine the performance of the pruned model quickly and accurately. The authors of the EagleEye pruning method believe that the root cause of the time-consuming fine-tuning part of the traditional pruning process is the large impact of the BN (batch normalization) layer of the network on the model accuracy. The parameters of the BN layer are fine-tuned multiple times to adapt the pruned model to improve the detection accuracy. The EagleEye pruning method uses Adaptive BN to continuously modify the parameters of the BN layer in the pruned network model utilizing samples from the partial training set and obtains a network model with the best pruning effect. The EagleEye pruning process is shown in Fig. 5.

## RESULTS

In this section, ablation experiments were carried out to confirm the effectiveness of the improvement proposed in this article. Then, the RCID-YOLOv5s method was compared with Faster R-CNN (*Ren et al., 2017*), YOLOv3 (*Redmon & Farhadi, 2018*), and YOLOv5s (*Glenn, 2020*). The experiments were conducted on Intel Xeon Gold 5218 @2.3 GHz CPU,

**Table 1  The initialization parameters of the network model.**

| Input size | Batch size | Epoch | Momentum | Learning rate | Optimizer |
|---|---|---|---|---|---|
| 608×608 | 16 | 300 | 0.9 | 0.01 | Adam |

TITAN RTX GPU, 12GB RAM, and Windows 10, and the framework for deep learning was PyTorch 1.9 in Python 3.7. The initialization parameters of the network model were shown in Table 1.

## Evaluation metric

This article used average precision (AP) as the evaluation metric (*Tong, Wu & Zhou, 2020*). AP refers to the size of the area enclosed by the precision-recall curve. The higher value of the AP, the better the overall performance of the network model. The specific formulas for precision, recall, and average precision are as follows:

$$P = \frac{TP}{TP + FP}, \tag{1}$$

$$R = \frac{TP}{TP + FN} \tag{2}$$

$$AP = \int_0^1 P(R)\mathrm{d}R \tag{3}$$

where TP (true positive) indicates the samples after network model detection and classification match the labeled samples, FP (false positive) indicates that the samples after network model detection and classification are not included in the labeled samples, and FN (false negative) indicates that the labeled samples are not detected.

## Dataset pre-processing

There is a lack of public datasets suitable for deep learning training in the research of railway catenary insulator defects detection. The railway catenary insulator dataset used in this article was photographed by inspection vehicles on the Zhengzhou-Xuzhou High-speed Railway, with a total of 1,300 images. In order to enrich the number of defective insulator images in the dataset and improve the generalization ability of the network model, the data enhancement operations such as random rotation were carried out on the original dataset to obtain more defective insulator images in different locations. The images after data augmentation are shown in Fig. 6. The final dataset consisted of 2,000 annotated images, and these images were annotated using the LabelImg (*Tzutalin, 2017*) annotation tool. The dataset was split into 70% for training, 10%for validating and 20% for testing.

## Ablation experiment

The CSP module is mainly used for feature extraction in the backbone, so the TA module is embedded after the CSP module of the backbone to give more weight to the effective

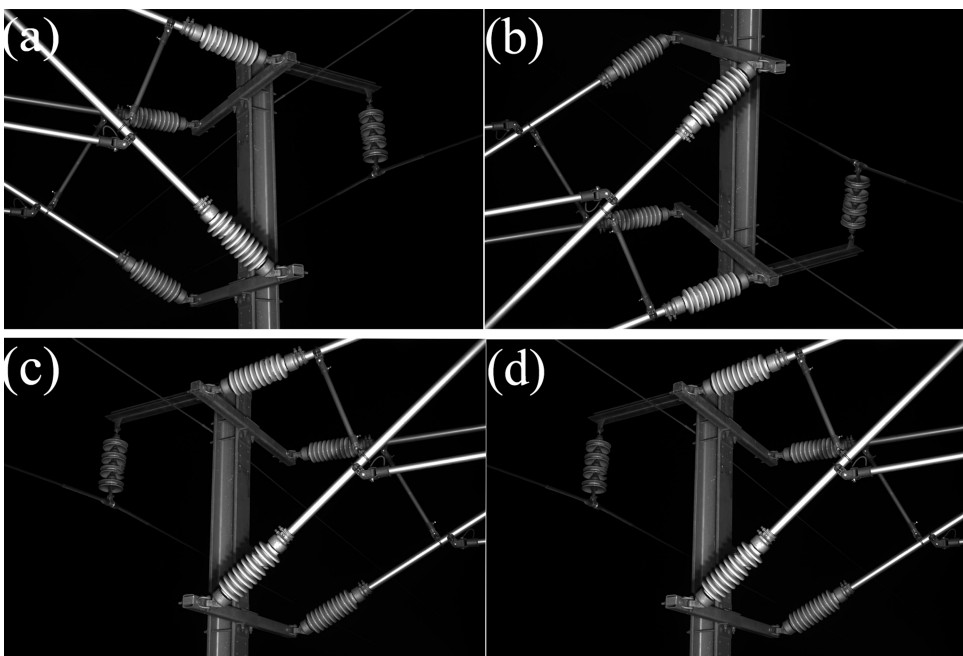

**Figure 6** **The images after data augmentation.** (A) Original image, (B) vertical flip, (C) horizontal flip, (D) shift scale rotate.

feature information. As shown in Fig. 2, the embedding position of the TA modules can be chosen among four positions, 1, 2, 3, and 4. The performance of network models with TA modules embedded in different locations was compared and analyzed to select the network model with the best performance for defect detection of railway catenary insulators.

As shown in Table 2, embedding the TA module in the network model can improve the detection accuracy, but this operation increases the parameters and the detection time of the network model. As the number of embedded TA modules increases, the size of the network model also increases, which leads to overfitting and reduce detection accuracy of the network model. Therefore, the network model that achieves the optimal balance between detection speed and accuracy should be selected. As can be seen from Table 2, the network model with the TA modules added at position 1 and position 4 has the best overall results. Specifically, the AP value of the network model is 99.1%, the detection speed of the network model is 0.052 s per image, and the model size of the network model is only 15.9MB. Therefore, the TA modules in this article were embedded in positions 1 and position 4 of the network model, as shown in Fig. 2.

The results of the ablation experiment are shown in Table 3. Giga floating point operations (GFLOPs) and parameters are used in the experiments to reflect the complexity and parameters of the network model. From Table 3, it can be seen that adding TA modules or adding small object detection layers can improve the performance of the network model. The experimental results in Table 3 show that embedding the TA modules in the network model can improve the AP value of the network model by 5.1%; adding the SODL can

**Table 2  The experimental results of the TA module added in different locations.**

| 1 | 2 | 3 | 4 | AP | Detection time (s/image) | Model size (MB) |
|---|---|---|---|---|---|---|
| √ | | | | 0.984 | 0.049 | 15.8 |
| | √ | | | 0.989 | 0.052 | 15.9 |
| | | √ | | 0.986 | 0.054 | 16.2 |
| | | | √ | 0.978 | 0.052 | 15.9 |
| √ | √ | | | 0.989 | 0.057 | 15.9 |
| √ | | √ | | 0.992 | 0.059 | 16.2 |
| √ | | | √ | 0.991 | 0.052 | 15.9 |
| | √ | √ | | 0.984 | 0.061 | 16.3 |
| | √ | | √ | 0.988 | 0.055 | 16.0 |
| | | √ | √ | 0.991 | 0.054 | 16.3 |
| √ | √ | √ | | 0.980 | 0.062 | 16.3 |
| √ | √ | | √ | 0.985 | 0.058 | 16.1 |
| √ | | √ | √ | 0.987 | 0.060 | 16.4 |
| | √ | √ | √ | 0.988 | 0.061 | 16.5 |
| √ | √ | √ | √ | 0.985 | 0.063 | 16.5 |

**Table 3  Ablation experiment results.**

| TA | SODL | Prune | AP | GFLOPs | Parameters | Detection time(s/image) | Model size (MB) |
|---|---|---|---|---|---|---|---|
| | | | 0.913 | 16.3 | 7059304 | 0.031 | 13.7 |
| √ | | | 0.964 | 16.1 | 7090720 | 0.034 | 13.7 |
| | √ | | 0.980 | 30.6 | 8075008 | 0.049 | 15.8 |
| √ | √ | | 0.991 | 31.0 | 8145880 | 0.052 | 15.9 |
| √ | √ | √ | 0.980 | 18.1 | 4447369 | 0.038 | 9.28 |

increase the AP value of the network model by 6.7%; embedding both the TA modules and SODL in the network model can increase the AP value of the network model by 7.8%.

However, the embedding of the TA modules and SODL leads to an increase in the computational complexity of the network model, which results in slower detection speed. To overcome this problem, model pruning is used to remove unimportant parameters or layers in the network model to improve the detection precision and speed. The results in Table 3 show that the AP value of the pruned network model reaches 98.0%, which is 6.7% higher than the original network model, and the parameters of the pruned network model is minimum. Meanwhile, the detection speed of the pruned network model is improved compared to the network model with the introduction of TA modules and SODL. Therefore, the final network model has the best overall performance and achieves a good balance between detection speed and precision.

**Table 4   The results of different models.**

| Method | AP | GFLOPs | Parameters | Detection time (s/image) | Model size (MB) |
|---|---|---|---|---|---|
| Faster R-CNN | 0.966 | / | / | 1.106 | 1034.2 |
| YOLOv3 | 0.962 | 154.9 | 61508200 | 0.135 | 117 |
| YOLOv5s | 0.913 | 16.3 | 7059304 | 0.031 | 13.7 |
| RCID-YOLOv5s | 0.980 | 18.1 | 4447369 | 0.038 | 9.28 |

## Comparison experiment

The RCID-YOLOv5s model was further compared with the Faster R-CNN (*Ren et al., 2017*), YOLOv3 (*Redmon & Farhadi, 2018*), and YOLOv5s. The results of different models are shown in Table 4. It can be seen from Table 4 that the RCID-YOLOv5s has the highest AP value of 98.0%, and the RCID-YOLOv5s also have the smallest number of parameters and model size. In terms of the addition of TA modules and a small object detection layer, the improved network model can give more weight to important features as well as generate a larger feature map. Thus, the ability of the network model to detect and locate small objects can be improved. However, the complexity of the network model is higher and the detection speed is slower than the original network model. Overall, it seems that RCID-YOLOv5s is optimal for the detection of defects in railway catenary insulators.

In Fig. 7, the defect detection results of railway catenary insulators by different methods are given. Figure 7A shows the original annotated image, Fig. 7B shows the detection results of the Faster R-CNN network model, Fig. 7C shows the detection results of the YOLOv3 network model, Fig. 7D shows the detection results of the YOLOv5s network model, and Fig. 7E shows the detection results of the RCID-YOLOv5s network model. Comparing the detection results of different network models, it can be seen that only RCID-YOLOv5s can accurately identify and locate the defects of insulators in railway catenary, Faster R-CNN, YOLOv3, and YOLOv5s all have missed and false detection, which is due to the defects of insulators are too small and the network model is not easy to extract features. Therefore, the RCID-YOLOv5s proposed in this article can accurately identify and locate the defects of railway catenary insulators.

## CONCLUSION

In this work, the RCID-YOLOv5s is proposed for railway catenary insulator defects detection. By adding an attention mechanism module to the network model to improve the weight assignment of important features; a small object detection layer is added to the network model to capture more shallow feature information of small objects. To solve the problem of increasing network parameters caused by increasing the number of network layers, the method of model pruning is used to remove some unimportant parameters from the network model, which reduces the number of parameters in the network model and improves the detection speed of the network model. Compared with the original YOLOv5s model, the value of AP is increased by 6.7%, reaching 98.0%, and has a lower miss detection

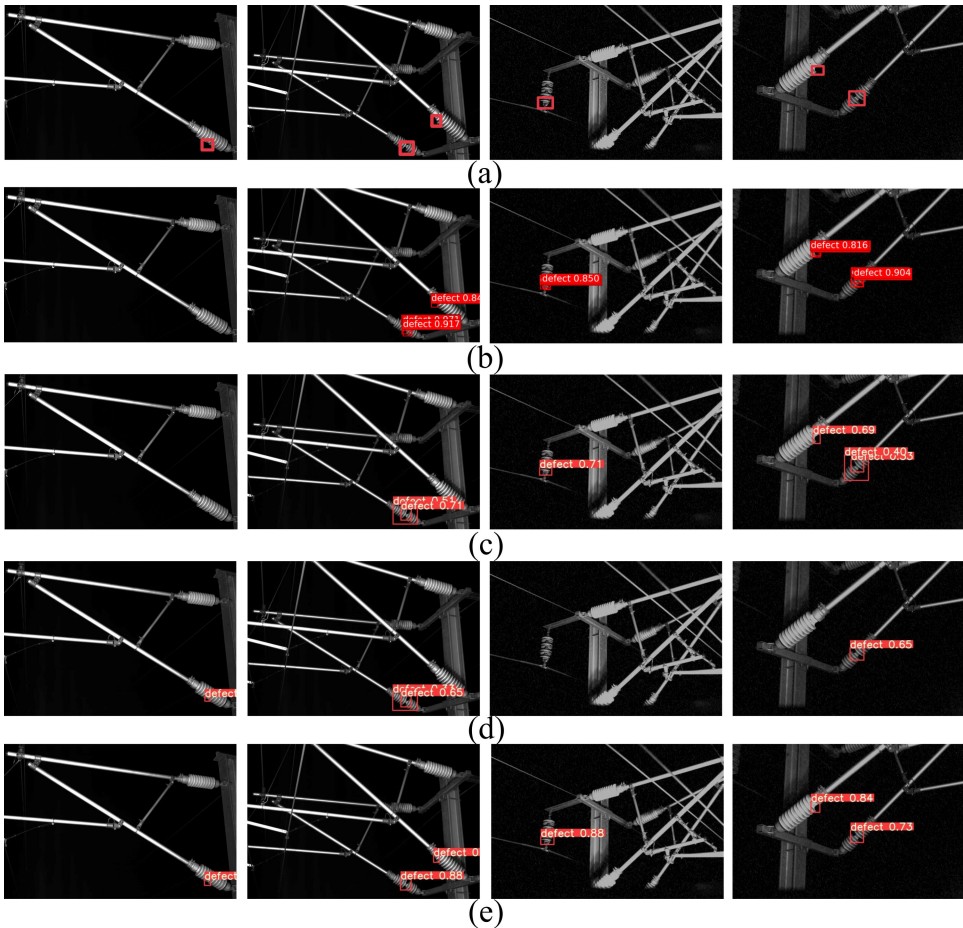

**Figure 7   The defect detection results of railway catenary insulators for different methods.** (A) Original annotated image, (B) Faster R-CNN, (C) YOLOv3, (D) YOLOv5s, (E) RCID-YOLOv5s.

rate. Therefore, defects in railway catenary insulators can be detected quickly and accurately by the RCID-YOLOv5s.

Due to the limitation of the number of original images, the generalization ability of the network model still has space to improve. The insulator defect types contained in the insulator image dataset need to be continuously collected and improved shortly research to further enhance the generalization capability of the network model. Therefore, we will further improve the detection speed of the network model without affecting the detection accuracy.

### Funding

This work was supported by the Special Project of Central Government for Local Science and Technology Development of Hubei Province No. 2019ZYYD020, the Natural Science

Foundation of Hubei Province No. 2022CFA007, and the Hubei University of Technology Ph. D. Research Startup Fund Project No. BSQD2020014. The funders had no role in study design, data collection and analysis, decision to publish, or preparation of the manuscript.

### Grant Disclosures

The following grant information was disclosed by the authors:
Special Project of Central Government: 2019ZYYD020.
Natural Science Foundation of Hubei Province: 2022CFA007.
Hubei University of Technology Ph. D. Research Startup Fund Project: BSQD2020014.

### Competing Interests

The authors declare there are no competing interests.

### Author Contributions

- Jing Tang conceived and designed the experiments, prepared figures and/or tables, and approved the final draft.
- Minghui Yu performed the experiments, performed the computation work, prepared figures and/or tables, and approved the final draft.
- Minghu Wu conceived and designed the experiments, analyzed the data, authored or reviewed drafts of the article, and approved the final draft.

### Data Availability

 The code is available in the Supplemental Files.

### Supplemental Information

Supplemental information for this article can be found online at http://dx.doi.org/10.7717/peerj-cs.1474#supplemental-information.

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
