# Peer review of "Detection of railway catenary insulator defects based on improved YOLOv5s"

_PeerJ Computer Science, doi:10.7717/peerj-cs.1474_

## Round 0.1 · original submission · Major Revisions

The reviewers suggest that you need to improve the English in your manuscript. Additionally, they raise some concerns regarding the methodology and the results. If you address all of the suggestions provided, there is a possibility that your manuscript may be considered.

Reviewer 1 ·

Basic reporting

The use of English in the manuscript is generally adequate, however, there are some expressions that are too colloquial, some typos, and a few sentences that are eighter to complex or that have small typos. Some examples are:
- “The catenary is a kind of power supply line …”
- “Wei et al. proposed a method that local features are extracted by …”
- “Liu et al. to improve the detection accuracy for insulator defects by adding …”
The manuscript would benefit from a full revision of the text to make sure these and other small typos are corrected.

The manuscript adequately presents the problem and related works.
A few extra keywords may be beneficial to help with indexing the manuscript.

The way the figures are presented makes it hard to visualize the differences between the different methods tested. I suggest combining related images (e.g. Combine all YoloV3 images into a single .png file, same for the augmented images)
Figure 7a presents 4 catenary images that are later processed by different systems. The first image is different in Fig 7b and Fig 7c. Please, check this, as the insulators in this image does not seem to present any damage, which would correspond to a correct detection.
Fig 2. is hard to follow, authors should try to rearrange the different components of the image to make it easier for the reader to be able to understand the basic design of the neural network without even reading the paper. The use of different colors may be beneficial in this subject.

Experimental design

The experiments and methods are well described.
However, further discussion relating the training of the model should be made, some topics that should be addressed are:
- AP is used as the “network evaluation indicator”. This reviewer wonders why not other more common methods, such as IoU, are used instead. This is especially relevant in case of large images with small detectable objects, such as in this case.
- No information about the loss function nor optimizer are provided
- Authors provide the code of the model. However, this code is useless without a Readme file to indicate how to test it and the dataset.
- The manuscript state that the “final dataset contains 2000 images”, but it is not stated how many original images were available before data augmentation. This data is relevant to evaluate whether the original dataset is sufficiently representative of the problem.

The manuscript states that a larger feature map of size 152x152 pixels is used to improve the detection of small objects. Would it be possible to use even bigger feature maps? Would that help detection? Please, discuss on this.

Validity of the findings

The results presented in this manuscript cannot be contrasted, as the authors have not provided access to the dataset. I would suggest making the data and code openly available for other researchers to replicate the results.

The results presented by the authors are relevant and demonstrate a great performance in relation to other state of the art systems.

Additional comments

The manuscript "Detection of Railway Catenary Insulators Defects based on RCID-YOLOv5s" presents a novel system for the detection of defects or wear on catenary insulators, a type of equipment of railway systems.
The document presents a well-structured work which is relevant to the field of study.
However, this reviewer considers that some changes need to be made to the manuscript prior to its publication.

Cite this review as

Reviewer 2 ·

Basic reporting

• The English language should be improved. Certain words or phrases are frequently reiterated within the same sentence, leading to redundancy and lack of clarity.
“brace sleeve screws” on lines 71, 72 and 73.
“the detection speed of the network” on lines 196, 197 and 198.

• Authors need to check the numbering of tables and figures.

• To ensure clarity, it is advisable to leave a visible gap between an acronym and its corresponding definition. This helps the reader distinguish the acronym from the rest of the text and aids in comprehension. Authors should review some lines such as: 24, 69, 70, 129, 132, 166 and 226.

• On line 142 it would not be necessary to indicate the acronym Triplet Attention because it has already been previously defined in line 129.

• On line 233: “Where TP (Ture Positive) … “ Ture?

• Authors should check Supplemental file, a couple of images do not correspond to the manuscript.

Experimental design

• Could the authors provide more details on the dataset used in this study? What types of defects were included in the dataset? How were the images obtained? How many images were included in the dataset before data augmentation? 9 images appear in the supplemental file, were any more used?

Validity of the findings

• I suggest the authors restructure the end of the article as the conclusion section is missing. An alternative could be that 3.4 Comparison experiments section becomes the Discussion section. And the current Discussion section, with some minor revisions, could become the Conclusions section. In addition, it would be desirable for the authors to add future trends in their research.

• Has the RCID-YOLOv5s model been tested in a real railway inspection scenario? It would be important to verify the behavior of the model in real situations different from the original images.

• Have the authors considered implementing this model in real-time detection of railway catenary insulator defects?

Cite this review as

---

## Round 0.2 · Minor Revisions

The decision made in the manuscript is for minor revisions.

Pay special attention to the comments made by reviewer 1 and address these in the next version of the manuscript.

Reviewer 1 ·

Basic reporting

The use of English in the manuscript has been clearly improved. However, there are still some errors:
-Incorrect use of horizontal tab in Line 38
-Excesive use of the word "object" in Lines 178 and 179
-"...experiments are using..." in Line 215
-Language level in subsections 3.2 and 3.3 is low. Moreover, section 3.3 is hard to follow and seems to repeat the same information over and over again

Images have been improved and clarified, thank you for that.
Consider including vectorized images in your manuscript, as Image 2 may be hard to read when zoomed in.

In Line 41 it is stated that "manual inspection is time-consuming which implies that accuracy can not be guaranteed". Why?

Experimental design

The authors have addressed all my comments regarding this section.

Validity of the findings

The authors have addressed all my comments regarding this section.

Additional comments

This reviewer thanks the authors for their response to the first revision and the changes that have been made to the manuscript.

This reviewer considers that the paper may be considered for its publication after some minor corrections are made. Those potential corrections have been indicated to the authors

Cite this review as

Reviewer 2 ·

Basic reporting

The authors have addressed all the issues. No more comments.

Experimental design

No comment

Validity of the findings

No comment

Cite this review as

---

## Round 0.3 · accepted · Accept

Congratulations on this publication!

Reviewer 1 ·

Basic reporting

Authors have addressed all my comments and have modified the manuscript accordingly.
I have no further comments

This reviewer considers the manuscript should be considered for its publication in its current form

Experimental design

I have no further comments

Validity of the findings

I have no further comments

Additional comments

I have no further comments

Cite this review as